# Lucanthone, Autophagy Inhibitor, Enhances the Apoptotic Effects of TRAIL through miR-216a-5p-Mediated DR5 Upregulation and DUB3-Mediated Mcl-1 Downregulation

**DOI:** 10.3390/ijms23010017

**Published:** 2021-12-21

**Authors:** Ji Yun Yoon, Seon Min Woo, Seung Un Seo, So Rae Song, Seul Gi Lee, Taeg Kyu Kwon

**Affiliations:** 1Department of Immunology, School of Medicine, Keimyung University, Daegu 42601, Korea; libra3009@naver.com (J.Y.Y.); woosm724@gmail.com (S.M.W.); sbr2010@hanmail.net (S.U.S.); ssr02067@naver.com (S.R.S.); lsg100479@naver.com (S.G.L.); 2Center for Forensic Pharmaceutical Science, College of Pharmacy, Keimyung University, Daegu 42601, Korea

**Keywords:** lucanthone, DR5, miR-216a-5p, Mcl-1, DUB3

## Abstract

A lucanthone, one of the family of thioxanthenones, has been reported for its inhibitory effects of apurinic endonuclease-1 and autophagy. In this study, we investigated whether lucanthone could enhance tumor necrosis factor-related apoptosis-inducing ligand (TRAIL)-induced apoptosis in various cancer cells. Combined treatment with lucanthone and TRAIL significantly induced apoptosis in human renal carcinoma (Caki and ACHN), prostate carcinoma (PC3), and lung carcinoma (A549) cells. However, combined treatment did not induce apoptosis in normal mouse kidney cells (TCMK-1) and normal human skin fibroblast (HSF). Lucanthone downregulated protein expression of deubiquitinase DUB3, and a decreased expression level of *DUB3* markedly led to enhance TRAIL-induced apoptosis. Ectopic expression of DUB3 inhibited combined treatment with lucanthone and TRAIL-induced apoptosis. Moreover, lucanthone increased expression level of *DR5* mRNA via downregulation of *miR-216a-5p*. Transfection of *miR-216a-5p* mimics suppressed the lucanthone-induced DR5 upregulation. Taken together, these results provide the first evidence that lucanthone enhances TRAIL-induced apoptosis through DR5 upregulation by downregulation of *miR-216a-5p* and DUB3-dependent Mcl-1 downregulation in human renal carcinoma cells.

## 1. Introduction

Lucanthone is one of the family of thioxanthenones and is originally synthesized for use as anti-schistosomal drugs [1]. Lucanthone preferentially intercalates at AT-rich sequences in DNA, resulting in the inhibition of the dual-function base excision repair enzyme apurinic endonuclease-1 (APE1) [2]. In addition to demonstrating that treatment with lucanthone induces lysosomal membrane permeabilization (LMP), this study indicates that the function of lucanthone as an autophagy inhibitor is attributed to the induction of LMP [3]. Inhibition of autophagy is a new alternative in terms of enhancing the anticancer ability of anticancer drugs [4]. Therefore, many combined treatments are being developed with the aim of enhancing the activity of chemotherapeutic agents.

The tumor necrosis factor-related apoptosis-inducing ligand (TRAIL), a member of the TNF superfamily, targets the death receptors (DRs; DR4 and 5). Interaction with TRAIL and DRs allows for the recruitment of FAS-associated death domain and caspase-8, which results in the formation of the death-inducing signal complex, eventually inducing apoptosis [5]. Van Roosmalen et al. suggest the different preference to TRAIL according as DRs [6]. In case of DR5, because binding affinity to TRAIL is stronger at 37 °C compared to DR4, it can more trigger TRAIL-mediated apoptosis [7]. Because the expression levels of DR4 and/or DR5 are often higher in cancer cells than in normal cells, TRAIL has a high potential as a cancer therapeutic agent [8,9]. Although TRAIL can be effective in cancer cells, some cancer cells are resistant to TRAIL due to, e.g., high expression of anti-apoptotic proteins and constitutive activation of cell survival signals. Therefore, many studies are being conducted on the development of a novel TRAIL sensitizer to overcome resistance to TRAIL [10,11].

Mcl-1 is a one of Bcl-2 family proteins and possesses anti-apoptotic functions. Therefore, dysregulation of Mcl-1 enhances sensitization of anticancer drugs through mitochondria-dependent apoptosis, such as mitochondria outer membrane permeabilization and cytochrome *c* release from mitochondria [12]. Especially, downregulation of *Mcl-1* by pharmacological inhibitor or gene depletion overcome resistance to TRAIL in many cancer cells [13,14].

In our present study, we investigated the effect of lucanthone on TRAIL-mediated apoptosis in human renal carcinoma Caki cells. Lucanthone induced DUB3-dependent Mcl-1 downregulation and *miR-216a-5p*-mediated DR5 upregulation, resulted in the enhancement to TRAIL sensitivity in cancer. These results provide the evidence that lucanthone could be a novel sensitizer of TRAIL-resistant cancer cells.

## 2. Results

### 2.1. Lucanthone Enhances TRAIL Sensitivity in Human Renal Carcinoma Cells 

Lucanthone exerts a novel autophagy inhibitor that induces cathepsin D-mediated apoptosis [15,16]. We examined whether lucanthone enhances TRAIL-induced apoptosis in human renal carcinoma cells. First, we surveyed subtoxic concentration of lucanthone using cell viability assay and a chosen concentration less than 5 μM (Appendix A). As shown in Figure 1B, combined treatment with lucanthone and TRAIL markedly increased the sub-G1 population and cleavage of Poly (ADP-ribose) polymerase (PARP), which is the useful hallmark of apoptosis, in renal carcinoma cells (Caki-1 and ACHN). Because each lucanthone and TRAIL single treatment showed 15% and 10% of sub-G1 population in Caki cells, each treatment alone did not increase apoptosis (Figure 1B). Combined treatment with lucanthone and TRAIL induced typical apoptotic properties, including exemplary chromatin damage in the nuclei and DNA fragmentation in Caki cells (Figure 1C,D). Additionally, combined treatment increased caspase-3 (DEVDase) activity (Figure 1E). To prove the involvement of caspase-dependent apoptosis by lucanthone with TRAIL, we examined effect of pan-caspase inhibitor (z-VAD-fmk (z-VAD)). z-VAD completely abolished apoptosis and cleavage of caspase-3 by combined treatment with lucanthone plus TRAIL (Figure 1F). Therefore, these results suggest that lucanthone enhances to TRAIL-induced apoptotic cell death in cancer cells.

### 2.2. Effect of Lucanthone on Expression Levels of Apoptosis-Related Proteins

To investigate how lucanthone enhances to TRAIL-mediated apoptosis, Western blot was performed to examine the expression levels of apoptosis-related proteins in various concentrations of lucanthone-treated Caki and ACHN cells. Lucanthone markedly decreased Mcl-1 expression and increased DR5 expression, while other apoptosis-related proteins (IAP and Bcl-2 family proteins) were not altered by lucanthone treatment (Figure 2A). To investigate the critical role of the Mcl-1 downregulation in the sensitizing effect of lucanthone on TRAIL-mediated apoptosis, we applied Caki cells overexpressing Mcl-1 (Caki/Mcl-1). Ectopic expression of Mcl-1 effectively blocked combined treatment-induced apoptosis and PARP cleavage (Figure 2B). Survivin was decreased by lucanthone treatment. However, overexpression of survivin did not inhibit combined treatment-induced apoptosis (Figure 2C). Thus, these data indicate that lucanthone-mediated Mcl-1 downregulation is associated with the enhancement of TRAIL-induced apoptosis.

### 2.3. Lucanthone Decreases Mcl-1 Protein Expression through Proteasome–Ubiquitin Pathway

Lucanthone treatment induced downregulation of Mcl-1 in a time-dependent manner in Caki cells (Figure 3A). To investigate the molecular mechanism of lucanthone-induced Mcl-1 downregulation, we examined the *Mcl-1* mRNA expression in lucanthone-treated cells. The RT-PCR and qPCR results showed that lucanthone does not modulate *Mcl-1* expression at the transcriptional level (Figure 3B). Therefore, we reasoned that lucanthone may regulate Mcl-1 at the post-translational level. First, we investigated stabilization of Mcl-1 protein using cycloheximide, a protein biosynthesis inhibitor. Expectably, lucanthone decreased the protein stability of Mcl-1 in a CHX chase assay (Figure 3C). Next, the proteasome inhibitors (MG132 and lactacystin) recovered lucanthone-induced Mcl-1 downregulation (Figure 3D). Taken together, our results suggest that lucanthone induces downregulation of Mcl-1 through the proteasome–ubiquitin pathway.

### 2.4. Lucanthone-Mediated DUB3 Downregulation Plays a Critical Role in Mcl-1 Degradation and TRAIL Sensitization

Previous studies reported that deubiquitinases contribute to Mcl-1 deubiquitination and stability [17,18]. To evaluate the expression levels of deubiquitinase, which are involved in regulation of Mcl-1 stability, we initially analyzed various deubiquitinase expression levels in lucanthone-treated cells. Lucanthone induced the reduction of DUB3 expression in a dose-dependent manner but had no effect on E3 ligases (Cdc20 and β-TrCP) and deubiquitinases (USP9X and OTUD1) expression in Caki and ACHN cells (Figure 4A). Reduced expression of deubiquitinase is likely related to lucanthone-mediated degradation of Mcl-1 protein. Therefore, we focused on DUB3 in lucanthone-induced Mcl-1 downregulation. Next, we examined the ubiquitination of Mcl-1 by lucanthone and found that lucanthone directly ubiquitylases Mcl-1 (Figure 4B). Next, we investigated the importance of the DUB3 downregulation in TRAIL-induced apoptosis, Caki cells were transiently transfected with *DUB3* siRNA and then treated TRAIL. Knockdown of *DUB3* markedly increased TRAIL-induced apoptosis and reduced Mcl-1 expression (Figure 4C). Furthermore, overexpression of DUB3 effectively inhibited combined treatment lucanthone and TRAIL-induced apoptosis and Mcl-1 downregulation (Figure 4D). These data demonstrate that DUB3 is important role in stability of Mcl-1 and sensitization of TRAIL by lucanthone.

### 2.5. Lucanthone Increases DR5 mRNA Expression via miR-216a-5p Downregulation

DRs play a critical role in TRAIL-mediated apoptosis [19]. Lucanthone upregulated DR5 protein expression in a concentration-dependent manner but had no effect on DR4 protein expression (Figure 2A). In addition, lucanthone only increased DR5 expression on cell surface, but not DR4 (Figure 5A and Appendix A). To investigate the lucanthone-mediated DR5 upregulation at the transcriptional levels, we examined *DR5* mRNA level promoter activity. Interestingly, lucanthone increased the *DR5* mRNA level, but not the *DR5* promoter activity (Figure 5B,C). Next, we examined the impact of lucanthone on *DR5* mRNA stability using the actinomycin D (Act D). Co-treatment with Act D and lucanthone better maintained the *DR5* mRNA level compared to Act D alone (Figure 5D). Next, we investigated involvement of microRNA in lucanthone-mediated *DR5* mRNA stabilization. As a result of bioinformatic TargetScan analysis (http://www.targetscan.org/, accessed on 8 November 21), it was putatively identified that four microRNAs (*miR-7-5p*, *miR-21-3p*, *miR-221-3p*, and *miR-216a-5p*) are involved in DR5 mRNA stabilization [20]. As a result of confirming the relevance of the four miRNAs, *miR-216a-5p* mimics only suppressed the lucanthone-induced DR5 upregulation (Figure 5E and Appendix A). To verify the involvement of *miR-216a-5p* in lucanthone-induced TRAIL sensitization, we examined apoptosis by combinations of lucanthone and TRAIL after transfection of *miR-216a-5p* mimics. The *miR-216a-5p* mimics partially inhibited lucanthone plus TRAIL-induced apoptosis (Figure 5F). In Caki and PC3 cells, *miR-216a-5p* expression was decreased by lucanthone treatment in a concentration-dependent manner (Figure 5G). Thus, these data indicate that downregulation of *miR-216a-5p* by lucanthone has a role in upregulation of DR5 and TRAIL sensitization.

### 2.6. Lucanthone Plus TRAIL Treatment Effects on Apoptosis in Various Cancer Cells, But Not in Normal Cells

We further examine whether lucanthone plus TRAIL induces apoptosis in other carcinoma cells and normal cells. As shown in Figure 6A, co-treatment with lucanthone and TRAIL increased the sub-G1 population and PARP cleavage in prostate cancer (PC3) and human lung cancer (A549) cells. Moreover, lucanthone upregulated DR5 and downregulated DUB3 and Mcl-1 in both cancer cells (Figure 6B). However, combined treatment had no effect on morphological apoptotic bodies and sub-G1 populations in normal human skin fibroblast (HSF) and TCMK-1 cells (Figure 6C). Therefore, these data indicate that lucanthone sensitizes to TRAIL-induced apoptotic cell death in cancer cells.

## 3. Discussion

In this study, we demonstrated that autophagy inhibitor, lucanthone, enhanced TRAIL-induced apoptotic cell death in cancer cells, but not in normal cells. We found that *miR-216a-5p*-mediated DR5 upregulation and DUB3-dependent Mcl-1 downregulation play a crucial role in lucanthone-enhanced sensitization of TRAIL-mediated apoptosis. Downregulation of DUB3 deubiquitinase was associated with Mcl-1 downregulation in lucanthone-treated cells. In addition, lucanthone upregulated DR5 through the downregulation of *miR-216a-5p* expression. Therefore, we demonstrated that lucanthone could enhance TRAIL-induced apoptotic cell death through the modulation of Mcl-1 and DR5 expression. 

Lucanthone, a novel autophagic inhibitor, induces apoptosis via cathepsin D accumulation and enhances histone deacetylase inhibitor (vorinostat)-induced cell death in MDA-MB-231 breast cancer cells [3]. Lucanthone also inhibits the endonuclease activity of APE1 by direct protein binding without affecting its redox activity [2]. Several studies have shown that lucanthone is currently being investigated as a sensitizer to chemotherapy and radiation due to APE inhibitory effects [2,21]. Recently, the combined therapy between autophagic inhibitors and conventional anticancer drugs have been investigated in clinical trials [22,23,24]. However, when autophagy inhibitors are combined with anticancer drugs, various problems exist as to whether to use them for specific cancers. 

Mcl-1 is an antiapoptotic member of the Bcl-2 family that has been found to play a critical role in the survival of various types of normal cells and resistance of tumor cells to various anticancer drugs [25]. Mcl-1 expression is modulated by transcriptional, post-transcriptional, translational, and post-translational levels [25]. As shown in Figure 3B, *Mcl-1* mRNA levels were not changed by lucanthone treatment. Multiple ubiquitin ligases and deubiquitinases modulate Mcl-1 stability through the ubiquitin–proteasome system pathway. To date, at least six E3 ligases (Mule, β-TrCP, FBW7, Cdc20, TRIM17, and FBXO4) have been reported that involved in ubiquitination of Mcl-1 [17]. As shown in Figure 4A, Mcl-1-targeting E3 ligases (Cdc20 and β-TrCP) and Mcl-1-targeting deubiquitinases (USP9X and OTUD1) were not altered by lucanthone. Interestingly, lucanthone only induced downregulation of DUB3 (Figure 4A). To confirm the involvement of DUB3 in lucanthone-enhanced TRAIL-mediated apoptosis, we knocked down or overexpressed DUB3. Knockdown of *DUB3* increased TRAIL-mediated apoptosis, whereas ectopic expression of DUB3 decreased combination of lucanthone plus TRAIL-induced apoptosis and Mcl-1 downregulation (Figure 4C,D). Wu et al. reported that DUB3 interacts with and deubiquitinates in Lys40 residue within the Mcl-1 [26]. PaTrin-2, a DNA repair protein O6-methylguanine-DNA methyltransferase inhibitor, suppresses DUB3 at the transcriptional level [26]. However, further studies on the molecular mechanism of DUB3 downregulation by lucanthone are needed. 

The expressional levels and functional modification of DR5 regulate sensitivity of TRAIL-induced apoptosis [27,28,29]. Lucanthone increased *DR5* mRNA stability (Figure 5C). MicroRNAs are small noncoding RNAs that negatively regulate the expression of multiple genes by translational repression [30]. Several studies have sought to identify the various microRNA targeting DRs of mRNA [20,31]. Above all, we focused on microRNAs that are involved in TRAIL resistance, such as *miR-7-5p*, *miR-21-3p*, *miR-221-3p*, and *miR-216a-5p* [32,33,34]. Previous studies have reported that knockdown of *miR-21-3p* increases TRAIL-induced apoptosis in glioma and liver cancer cells [33,35]. In our study, *DR5* mRNA and protein were upregulated by lucanthone, whereas expression of DR4 protein did not change (Figure 2A and Figure 5B). Therefore, we investigated the involvement of these microRNA in lucanthone-induced DR5 upregulation and TRAIL sensitivity. The three microRNA mimics (*miR-7-5p*, *miR-21-3p*, and *miR-221-3p*) did not suppress the lucanthone-induced DR5 upregulation (Appendix A). However, our data clearly indicated that *miR-216a-5p* mimics inhibited lucanthone-mediated DR5 upregulation in cancer cells (Figure 5E). Furthermore, combinations of lucanthone and TRAIL-induced apoptosis were slightly prevented by *miR-216a-5p* mimics, and lucanthone suppressed *miR-216a-5p* levels (Figure 5F,G). Therefore, we verified that lucanthone upregulates DR5 expression through suppression of *miR-216a-5p*, thereby increasing TRAIL sensitization in cancer cells.

Taken all together, we suggest that the ubiquitin–proteasome pathway-mediated Mcl-1 downregulation and *miR-216a-5p*-mediated DR5 upregulation have a critical role in autophagy inhibitor (lucanthone)-mediated sensitization of cancer cells to TRAIL-induced apoptosis.

## 4. Materials and Methods

### 4.1. Cell Cultures and Materials

Human renal carcinoma (Caki and ACHN), human prostate cancer (PC3), human lung cancer (A549), and TCMK-1 were procured from American Type Culture Collection (Manassas, VA, USA). Korea Cell Line Bank (Seoul, Korea) provided normal human skin fibroblasts (HSF) cells. These cells were cultured in appropriate medium containing 10% fetal bovine serum, 1% penicillin/streptomycin, and 100 μg/mL gentamicin at 37 °C in a humidified atmosphere with 5% CO_2_. Lucanthone was purchased from Selleckchem (Houston, TX, USA). Human recombinant TRAIL and zVAD-fmk were provided by R&D system (Minneapolis, MN, USA). Lactacystin was supplied from Enzo Life Sciences (Ann Arbor, MI, USA). Cycloheximide, actinomycin D, and MG132 were provided from Sigma Chemical Co. (St. Louis, MO, USA). The primary antibodies were obtained as follows: anti-PARP, anti-cleaved caspase-3, anti-Bcl-xL, anti-DR5, anti-cIAP1, anti-Bax, and anti-Mcl-1 from Cell Signaling Technology (Beverly, MA, USA); anti-Bim and anti-XIAP from BD Biosciences (San Jose, CA, USA); anti-Bcl-2, anti-cIAP2, and anti-β-TrCP from Santa Cruz Biotechnology (St. Louis, MO, USA); anti-survivin from R&D system (Minneapolis, MN, USA); anti-DR4 and anti-Cdc20 from Abcam (Cambridge, MA, USA); anti-USP9X from Abnova (Taipei City, Taiwan); anti-caspase-3 and anti-c-FLIP from Enzo Life Sciences (San Diego, CA, USA); anti-DUB3 from Thermo Fisher Scientific (Waltham, MA, USA); anti-OTUD1 from Atlas Antibodies (Bromma, Sweden); anti-USP9X from Abnova (Taipei City, Taiwan); anti-USP1 from Bethyl Laboratories (Montgomery, TX, USA); and anti-actin from Sigma Chemical Co. (St. Louis, MO, USA).

### 4.2. Flow Cytometry Analysis

To analyze apoptosis, cells were harvested and fixed with 95% ethanol at least 1 h at 4 °C. Then, cells were incubated in 1.12% sodium citrate buffer containing RNase at 37 °C for 30 min, added to 50 μg/mL propidium iodide, and analyzed using BD Accuri™ C6 flow cytometer (BD Biosciences, San Jose, CA, USA) [36].

### 4.3. Western Blotting

Cells were lysed in RIPA lysis buffer (20 mM HEPES and 0.5% Triton X-100, pH 7.6) and supernatant fractions were collected. Proteins were separated by SDS-PAGE and transferred to the nitrocellulose membranes (GE Healthcare Life Science, Pittsburgh, PO, USA), incubated with a specific antibody, and bands were detected using Immobilon Western Chemiluminescent HRP Substrate (EMD Millipore, Darmstadt, Germany) [37].

### 4.4. DAPI, DNA Fragmentation Assay, and Caspase Activity Assay

To investigate the nuclei condensation, cellular nuclei cells were stained with 300 nM 4′, 6′-diamidino-2-phenylindole solution (Roche, Mannheim, Germany), and we viewed fluorescence images using fluorescence microscopy (Carl Zeiss, Jena, Germany) [38]. To analyze DNA fragmentation, we used the death detection ELISA plus kit (Boehringer Mannheim, Indianapolis, IN, USA) according to the manufacturer’s recommendations. To measure DEVDase activity, cells were incubated with reaction buffer containing acetyl-Asp-Glu-Val-Asp p-nitroanilide (Ac-DEVD-pNA) substrate, as previously mentioned [39].

### 4.5. Reverse Transcription-Polymerase Chain Reaction (RT-PCR) and Quantitative PCR (qPCR)

To isolate the total RNA, we used TriZol reagent (Life Technologies, Gaithersburg, MD, USA), and obtained cDNA using M-MLV reverse transcriptase (Gibco-BRL, Gaithersburg, MD, USA). For PCR, we used Blend Taq DNA polymerase (Toyobo, Osaka, Japan) with primers targeting *Mcl-1*, *DR5*, and *actin* as mentioned in our previous studies [40]. For qPCR we utilized SYBR Fast qPCR Mix (Takara Bio Inc., Shiga, Japan) and reactions were performed on Thermal Cycler Dice® Real-Time System III (Takara Bio Inc., Shiga, Japan). The following primers were used for the amplification of the target genes; *Mcl-1* (forward) 5′-ATG CTT CGG AAA CTG GAC AT-3′ and (reverse) 5′-TCC TGA TGC CAC CTT CTA GG-3′; *DR5* (forward) 5′- GAC CCT TGT GCT CGT TGT C-3′ and (reverse) 5′- TTG TTG GGT GAT CAG AGC AG-3′; and *actin* (forward) 5′-CTA CAA TGA GCT GCG TGT G-3′ and (reverse) 5′-TGG GGT GTT GAA GGT CTC-3′. We used actin as a reference gene to calculate the threshold cycle number (Ct) of DR5 gene and reported the delta-delta Ct values of the genes.

### 4.6. Luciferase Activity Assay

The DR5 (SacI) promoter constructs were transfected into the cells using Lipofectamine™2000 (Invitrogen, Carlsbad, CA, USA). Then, cells were collected and harvested in lysis buffer (25 mM Tris-phosphate pH 7.8, 2 mM EDTA, 1% Triton X-100, and 10% glycerol). The supernatants were used to measure the luciferase activity according to the manufacturer’s instructions (Promega, Madison, WI, USA).

### 4.7. Ubiquitination Assay

The assay was performed as described in our previous study [39]. Cells were transfected with HA-tagged ubiquitin (HA-Ub) and treated with lucanthone and MG132 for 12 h. Immunoprecipitation was performed using the anti-Mcl-1, and ubiquitination of endogenous Mcl-1 was checked using HRP-conjugated anti-Ub under denaturing conditions.

### 4.8. Transfection

For knockdown of genes, Caki cells were transfected with the control siRNA and *DUB3* siRNA (Bioneer, Daejeon, Korea) using Lipofectamine® RNAiMAX Reagent (Invitrogen, Carlsbad, California, USA). For overexpression of genes, Caki cells were transfected with pcDNA3.1(+) and pcDNA3.1(+)/DUB3 plasmids using Lipidofect-P (Lipidomia, Seongnam, Korea) for 24 h. Protein expression was checked by Western blotting.

### 4.9. Cell Surface Staining of DR5

Detached cells by 0.2% EDTA were washed with PBS, and then suspended in 100 μM PBS including 10% FCS and 1% sodium azide and added to the primary antibody (DR5-phycoerythrin, Abcam, Cambridge, MA, USA) for 1 h at room temperature. Then, the cells were washed with PBS including 10% FCS and 1% sodium azide and were suspended in 400 μL of PBS including 10% FCS and 1% sodium azide for the detection of DR5 expression on cell surface by flow cytometry.

### 4.10. Investigation of miR-216a-5p Expression Using qPCR

RT reactions were performed using a Mir-X™ miRNA First Strand Synthesis Kit (Takara Bio Inc., Shiga, Japan). For qPCR, we utilized SYBR Fast qPCR Mix (Takara Bio Inc., Shiga, Japan), and reactions were performed on Thermal Cycler Dice® Real Time System III (Takara Bio Inc., Shiga, Japan). The following primers were used for the amplification of the target genes; *miR-216a-5p* 5′-TAA TCT CAG CTG GCA ACT GTG A-3′.

### 4.11. Statistical Analysis 

The data were analyzed using a one-way ANOVA and post hoc comparisons (Student-Newman-Keuls) using the Statistical Package for Social Sciences 22.0 software (SPSS Inc.; Chicago, IL, USA).

## 5. Conclusions

Our study suggests that lucanthone, an autophagy inhibitor, sensitizes TRAIL-induced apoptosis via Mcl-1 downregulation and DR5 upregulation in cancer cells, but not normal cells. In mechanisms, lucanthone induces DUB3-dependent Mcl-1 downregulation and *miR-216a-5p*-mediated DR5 upregulation.

## Figures and Tables

**Figure 1 ijms-23-00017-f001:**
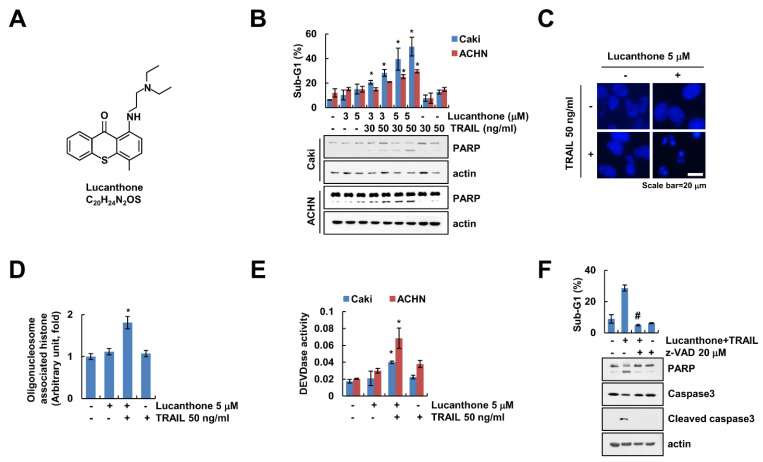
Lucanthone sensitizes TRAIL-induced apoptosis in human renal cancer cells. (**A**) Chemical structure of lucanthone (Formula: C_20_H_24_N_2_OS). (**B**) Caki and ACHN cells were treated with 3–5 μM lucanthone or/and 30–50 ng/mL TRAIL for 24 h; (**B**–**E**) Caki and ACHN cells were treated with 5 μM lucanthone or/and 50 ng/mL TRAIL for 24 h. (**C**) Condensation and (**D**) fragmentation of nuclei were examined using microscopy and DNA fragmentation assay kit, respectively. (**E**) DEVDase activity was examined using DEVDase substrate; (**F**) Caki cells were treated with 5 μM lucanthone and 50 ng/mL TRAIL for 24 h in the presence or absence of pretreatment with z-VAD for 30 min. The sub-G1 population and protein expression were measured by flow cytometry and Western blotting, respectively (**B**,**F**). The values in graph represent the mean ± SD of three independent samples. * *p* < 0.05 compared to the control. # *p* < 0.05 compared to the treatment of combinations of lucanthone and TRAIL.

**Figure 2 ijms-23-00017-f002:**
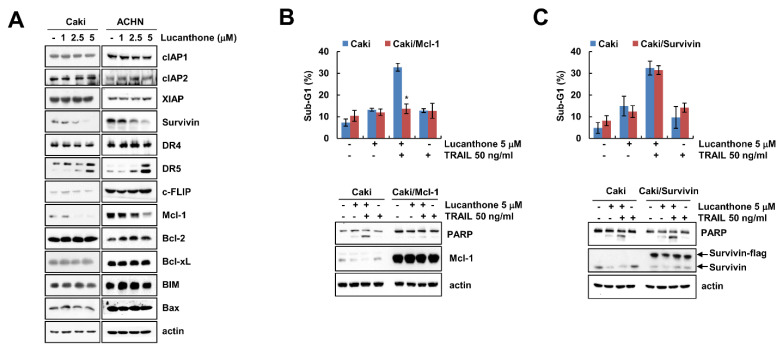
Lucanthone induces Mcl-1 downregulation and DR5 upregulation. (**A**) Caki and ACHN cells were treated with the 1-5 μM lucanthone for 24 h; (**B**,**C**) Caki and Mcl-1- or survivin-overexpressed Caki cells (**B**) Caki/Mcl-1 or (**C**) Caki/survivin) were treated with 5 μM lucanthone or/and 50 ng/mL TRAIL for 24 h. (**A**–**C**) The protein expression and (**B**,**C**) sub-G1 population were measured by Western blotting and flow cytometry, respectively. The values in graph represent the mean ± SD of three independent samples. * *p* < 0.05 compared to the combinations of lucanthone and TRAIL in Caki cells.

**Figure 3 ijms-23-00017-f003:**
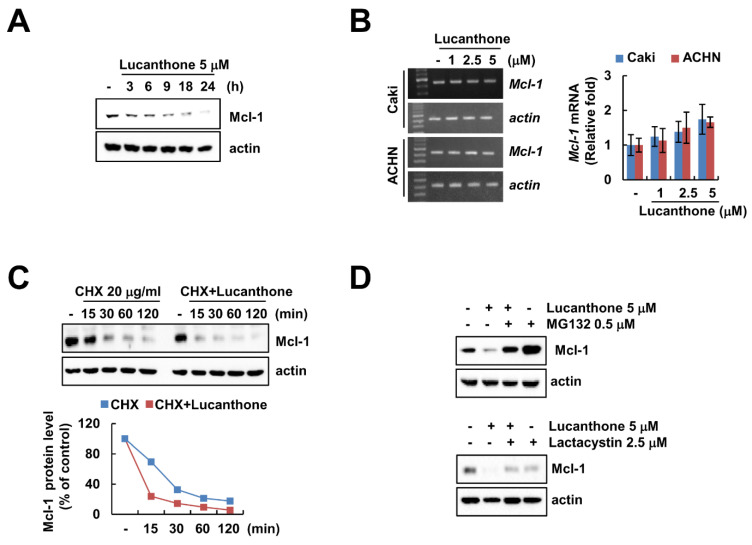
Lucanthone decreases Mcl-1 expression through the ubiquitin–proteasome pathway. (**A**) Caki and ACHN cells were treated with 5 μM lucanthone for the indicated time periods; (**B**) Caki and ACNH cells were treated with 1–5 μM lucanthone for 24 h. The mRNA levels of *Mcl-1* and *actin* were quantified by RT-PCR and qPCR; (**C**) Caki cells were pretreated with 20 μg/mL cycloheximide (CHX) for 30 min and then treated with 5 μM lucanthone for the indicated time periods. The band intensity was examined using Image J; (**D**) Caki and A549 cells were pretreated with 0.25 μM MG132 or 2.5 μM lactacystin for 30 min and then treated with 5 μM lucanthone for 24 h. The protein expression was measured by Western blotting. The values in graph represent the mean ± SD of three independent samples.

**Figure 4 ijms-23-00017-f004:**
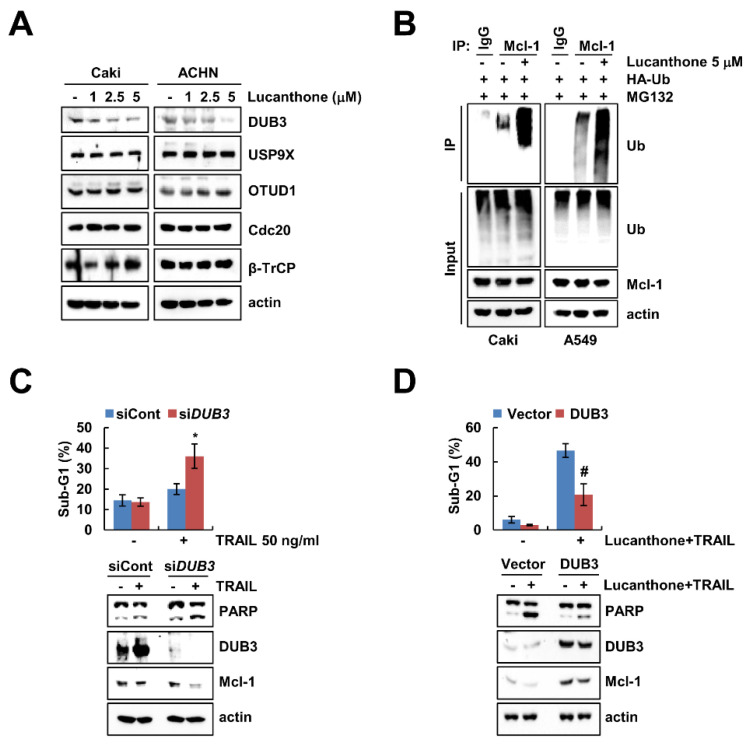
Lucanthone-induced DUB3 downregulation contributes to TRAIL sensitization. (**A**) Caki and ACHN cells were treated with the 1–5 μM lucanthone for 24 h; (**B**) Caki cells were transfected with HA-ubiquitin (HA-Ub) plasmid for 24 h. Then cells were treated with 0.5 μM MG132 and 5 μM lucanthone for 24 h. Mcl-1 ubiquitination was detected by Western blotting using HRP-conjugated anti-Ub antibody; (**C**) Caki cells were transfected control siRNA (siCont) or *DUB3* siRNA (si*DUB3*), and then treated with 50 ng/mL TRAIL for 24 h; (**D**) Caki cells were transfected with pcDNA3.1(+) (Vector) or pcDNA3.1(+)/DUB3 plasmids, and then treated with combinations of 5 μM lucanthone with 50 ng/ml TRAIL for 24 h. (**A**–**D**) The protein expression and (**C**,**D**) sub-G1 population were measured by Western blotting and flow cytometry, respectively. The values in graph represent the mean ± SD of three independent samples. * *p* < 0.05 compared to TRAIL in siCont-transfected cells. # *p* < 0.05 compared to the combinations of lucanthone and TRAIL in vector-transfected cells.

**Figure 5 ijms-23-00017-f005:**
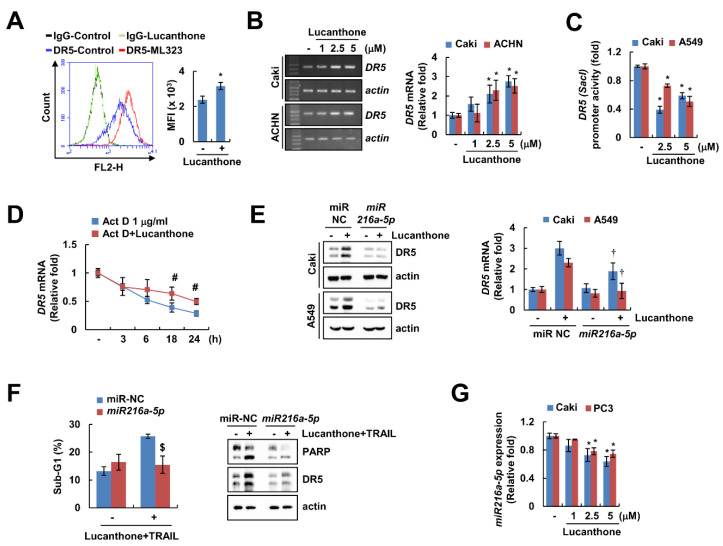
Lucanthone increases *DR5* mRNA expression through decrease of *miR-216a-5p*. (**A**) Caki cells were treated with 5 μM lucanthone for 24 h and analyzed DR5 expression on the cell surface using flow cytometry. (**B**) Caki and ACNH cells were treated with 1–5 μM lucanthone for 24 h. The mRNA levels of *DR5* and *actin* were examined by RT-PCR and qPCR; (**C**) Caki and A549 cells were transfected with *DR5 (-Sac1)* promoter and then treated with 2.5–5 μM lucanthone for 24 h. The cells were lysed, and then promoter activity was measured. (**D**) Caki cells were pretreated with 1 μg/mL actinomycin D (Act D) for 30 min and then treated with 5 μM lucanthone for the indicated time periods. The mRNA levels of *DR5* was quantified by RT-PCR and qPCR; (**E**,**F**) Caki and A549 cells were transfected with miR-negative control (NC) and *miR-216a-5p* and then treated with (**E**) 5 μM lucanthone or (**F**) combinations of 5 μM lucanthone with 50 ng/ml TRAIL for 24 h. (**G**) Caki and PC3 cells were treated with 1–5 μM lucanthone for 24 h. The expression of *miR-216a-5p* was determined by qPCR. (**E**,**F**) The protein expression and (**F**) sub-G1 population were measured by Western blotting and flow cytometry, respectively. The values in graph represent the mean ± SD of three independent samples. * *p* < 0.05 compared to control. # *p* < 0.05 compared to the Act D. † *p* < 0.05 compared to lucanthone in miR-NC. $ *p* < 0.05 compared to combinations of lucanthone and TRAIL in miR-NC.

**Figure 6 ijms-23-00017-f006:**
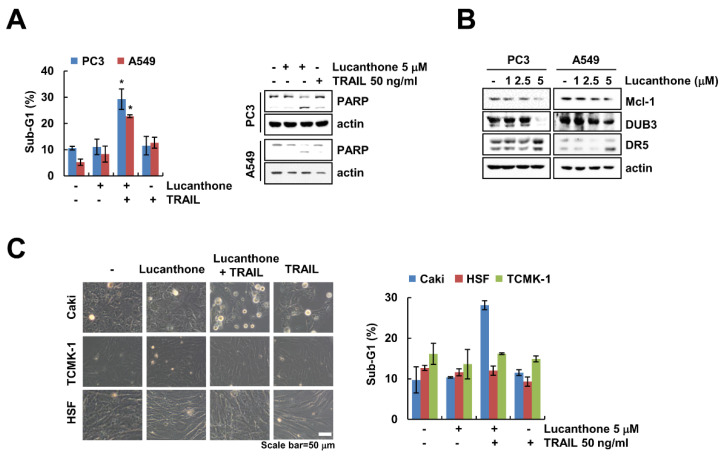
Effects of lucanthone on TRAIL sensitization in other cancer cells and normal cells. (**A**–**C**) Human cancer cell lines (prostate cancer; PC3 and lung cancer; A549) and normal cell lines (HSF and TCMK-1) were treated with (**A**,**C**) a combinations of 5 μM lucanthone with 50 ng/ml TRAIL or (**B**) 1–5 μM lucanthone for 24 h. (**A**,**C**) The sub-G1 population and (**A**,**B**) protein expression were measured by flow cytometry and Western blotting, respectively. (**C**) Cell morphology was imaged by interference light microscopy. The values in graph (**A**,**C**) represent the mean ± SD of three independent samples. * *p* < 0.05 compared to control.

## Data Availability

The data presented in this study are available on request from the corresponding author.

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
