# Peer review of "Lucanthone, Autophagy Inhibitor, Enhances the Apoptotic Effects of TRAIL through miR-216a-5p-Mediated DR5 Upregulation and DUB3-Mediated Mcl-1 Downregulation"

_ijms, 2021, doi:10.3390/ijms23010017_

Round 1
Reviewer 1 Report
The article “Lucanthone, Autophagy inhibitor, Enhances the Apoptotic Effects of TRAIL Through miR-216a-5p-Mediated DR5 Upregulation and DUB3-Mediated Mcl-1 Downregulation”, presented by Yoon and colleagues, provides a mechanistic explanation for how the autophagy inhibitor lucanthone augments cell death induction by the extrinsic pathway activator TRAIL. The manuscript is well written and technically sound but several minor concerns need to be addressed prior to publication of this work.
Minor concerns:
- It would be helpful to add the chemical structure of lucanthone to one of the figures.
- In the introductory paragraph, a brief expansion on the roles and functions of DR5 and Mcl-1 should be considered. This would give the readers not quite familiar with the players of the TNF family to better understand the reasoning for the experimental designs employed.
- Similarly, some of the strong claims stated in lines 41-45 should be adjusted a bit:
Because the expression levels of DR4 and/or DR5 are often/more frequently (choose one) higher in cancer cells than in normal cells, TRAIL has a high potential as a cancer therapeutic agent [5,6]. Although TRAIL can be effective in cancer cells, some cancer cells are resistant to TRAIL due to, e.g. high expression of anti-apoptotic proteins and constitutive activation of cell survival signals.
4. In the Results section, please expand on the reasoning (and explain the mechanism) for choosing sub-G1 population and PARP (spell out once) as surrogate markers of cell death (line 58).
5. Likewise, please explain in the text why cyclohexamide was used (line 104).
6. There was no mention where the normal human cell line HSF was obtained from.
Author Response
Could find attachment file?
Sincerely yours,
Taeg Kyu Kwon, Ph.D.

Reviewer 2 Report
The manuscript of Yoon et al. describes the possible antiapoptotic mechanism of action of lucantone, an autophagy inhibitor. The different biological assays are described and commented in detail and follow a rational order. The conclusions are based on the obtained results.
In my opinion there is only two questions.
1) In the text of the manuscript there are no cytotoxicity values of lucantone on the cell lines studied. These data could better clarify the real effect of lucantone on the results of cellular assays and could clearly highlight the effects of this drug. Furthermore, it would be appropriate to mention the criteria that led to the choice of drug concentrations used in the various biological assays.
2) In Materials and Methods, the flow cytometry analysis is described, but it is not indicated whether the method followed is present in bibliographic references or the manufacturer's instructions have been followed.
Provided that these issues are settled the paper is worthy of pubblication.
Author Response

(The authors gave the same response as above.)

Reviewer 3 Report
Dear authors, your manuscript presents interesting data showing for the first time that the topoisomerase inhibitor Miracil D or Lucanthone can induce an increase in TRAIL sensitivity in human cancer through ubiquitination and degradation of Mcl-1. Although the gain of function is, in my opinion rather modest, your data are sound; Yet some conclusions need to be strengthened.
First of all, you associate this gain of function with an increase in DR5 expression, one of the two TRAIL agonist receptors, and claim that this is due to Lucanthone-induced miR216-5p downregulation.
In order to sustain your conclusion that inhibition of miR216-5p induced by Lucanthone is playing a role in regulating DR5 expression levels and TRAIL sensitivity it would be important 1) to monitor DR5 membrane expression levels by flow cytometry and include too an analysis of DR4, since you solely rely on WB experiments and your anti_DR4 antibody is not the best for western blot analysis. 2) The effect of miR216-5p on DR5 mRNA and by extension Lucanthone on miR216-5p , although consistent are rather mild. How can you expect an effect on TRAIL-induced cell death without further demonstration ?. Could you impair or overexpress miR216-5p and see the effect on DR5 expression (mRNA / Membrane and TRAIL-induced apoptosis) ? 3) Most of your demonstration relies on Caki cells, could you extend your finding (Ubiquitination and degradation of Mcl-1 in other cells such as PC3 and A549, effect of miRNAs on DR5 expression) ? 4) You might also consider the possibility that Lucanthone may induce an ER Stress and through CHOP regulate DR5 expression level, regardless or miRNAs (one of your publication published in 2018 in Molecules). Please comment and test CHOP expression levels. 5) Survivin is also downregulated in your experimental model, yet this is not discussed in your manuscript. Why did you only consider Mcl1 ?
Last, given that the relationship with your miRNA is, so far, the weakest part of your manuscript I would urge you to strengthen the regulatory part (expression of the receptors) and negative regulation of Mcl1 in other cells in order to provide a strong message. You may also, should you like it, provide additional data to link and strengthen your findings with the miRNA.
Minor :
Figure 2 please show uncropped XIAp (ACHN). Why are there two bands in this cell line while there is only one in Caki ?
Figure 4 you claim that only DUB3 is inhibited by Lucanthone but your panel A shows that Cdc2 may also be affected. Please explain and correct in the text (tune down your statement that only DUB3 is affected).p8 line 221
Author Response

(The authors gave the same response as above.)

Round 2
Reviewer 3 Report
Many thanks for your clear answers. Could you please address the following points ?
#1 : The original blots provided in the revised version correspond to the first submission. Please provide an updated version.
#2 : It would also be important to provide a detailed update for the blots changed ie Fig 2 for XIAP and Fig 4 for CDC20. Not only the changed blot, but also a minimum set of important proteins assessed for the repeat should be shown (At least for the reviewer). For instance for Figure #2, you should consider showing SURVIVIN / DR5 / MCL1 and the actine in addition to the new XIAP BLOT. Same would apply for CDC2. In this case it would be important to show DUB3, USPX9, OTUD12, and bTcCP.
#3 Along the line, in DUB3 overexpressing CASKI cells, you could show also a higher exposure for DUB3, since the endogenous DUB3 is not detectable in mock transfected/infected cells panel D.
#4 Last you should probably check for cIAP2 blot in HCN (Fig 2) and USPX9 blot in Caski (Fig 4A) that differ in the manuscript as compared to the original blots. Please double check carefully all blots to provide the corresponding originals.
#5 How can you estimate the molecular weights in your western blot in the absence of marker ! Am I wrong ? If so please show the ladder.
Author Response

(The authors gave the same response as above.)

Round 3
Reviewer 3 Report
Many thanks for your clear answers. Could you please address the following points ?
#1 : The original blots provided in the revised version correspond to the first submission. Please provide an updated version.
#2 : It would also be important to provide a detailed update for the blots changed ie Fig 2 for XIAP and Fig 4 for CDC20. Not only the changed blot, but also a minimum set of important proteins assessed for the repeat should be shown (At least for the reviewer). For instance for Figure #2, you should consider showing SURVIVIN / DR5 / MCL1 and the actine in addition to the new XIAP BLOT. Same would apply for CDC2. In this case it would be important to show DUB3, USPX9, OTUD12, and bTcCP.
#3 Along the line, in DUB3 overexpressing CASKI cells, you could show also a higher exposure for DUB3, since the endogenous DUB3 is not detectable in mock transfected/infected cells panel D.
#4 Last you should probably check for cIAP2 blot in HCN (Fig 2) and USPX9 blot in Caski (Fig 4A) that differ in the manuscript as compared to the original blots. Please double check carefully all blots to provide the corresponding originals.
#5 How can you estimate the molecular weights in your western blot in the absence of marker ! Am I wrong ? If so please show the ladder.